# Learning Analytics on YouTube Educational Videos: Exploring Sentiment Analysis Methods and Topic Clustering

Ilias Chalkias, Katerina Tzafilkou *, Dimitrios Karapiperis * and Christos Tjortjis

School of Science & Technology, International Hellenic University, 14th km Thessaloniki, N. Moudania, 57001 Thessaloniki, Greece; ichalkias@ihu.edu.gr (I.C.); c.tjortjis@ihu.edu.gr (C.T.)
* Correspondence: ktzafilkou@ihu.edu.gr (K.T.); dkarapiperis@ihu.edu.gr (D.K.)

**Abstract:** The popularity of social media is continuously growing, as it endeavors to bridge the gap in communication between individuals. YouTube, one of the most well-known social media platforms with millions of users, stands out due to its remarkable ability to facilitate communication through the exchange of video content. Despite its primary purpose being entertainment, YouTube also offers individuals the valuable opportunity to learn from its vast array of educational content. The primary objective of this study is to explore the sentiments of YouTube learners by analyzing their comments on educational YouTube videos. A total of 167,987 comments were extracted and processed from educational YouTube channels through the YouTube Data API and Google Sheets. Lexicon-based sentiment analysis was conducted using two different methods, VADER and TextBlob, with the aim of detecting the prevailing sentiment. The sentiment analysis results revealed that the dominant sentiment expressed in the comments was neutral, followed by positive sentiment, while negative sentiment was the least common. VADER and TextBlob algorithms produced comparable results. Nevertheless, TextBlob yielded higher scores in both positive and negative sentiments, whereas VADER detected a greater number of neutral statements. Furthermore, the Latent Dirichlet Allocation (LDA) topic clustering outcomes shed light on various video attributes that potentially influence viewers' experiences. These attributes included animation, music, and the conveyed messages within the videos. These findings make a significant contribution to ongoing research efforts aimed at understanding the educational advantages of YouTube and discerning viewers' preferences regarding video components and educational topics.

**Keywords:** analytics; learning analytics; sentiment analysis; social media; topic clustering; YouTube comments; YouTube education



## 1. Introduction

Social media entail diverse modes of communication, collaboration, and the interactive expression of ideas [1]. The advent of social media platforms has profoundly transformed the global landscape, progressively supplanting conventional methods of communication, idea dissemination, and even individual learning approaches. Currently, social media is predominantly regarded as a means of communication and information exchange, prompting both enterprises and individuals to perceive it as a valuable marketing tool. While numerous studies have evaluated the impact of social media on the marketing research domain, this represents just one of its manifold potential applications. A captivating research area relating to social media is its contribution to the educational context and how it can influence the learning process. In their study, Zheng et al. [2] investigated the educational opportunities that arise from social media, particularly in fostering cooperative learning to enrich the learning experience. Notably, platforms like YouTube, which host video content, play a pivotal role in providing learners with easily accessible information devoid of constraints [3].

Numerous researchers support the idea that integrating social media in higher education teaching and learning practices can lead to bridging the gap between students [4,5].

Research has identified a correlation between the use of YouTube and its role in enhancing the learning process with one condition: the video must be relevant to the subject matter [6]. This study also highlights that incorporating visual elements, such as videos, supports the educational process as students become more engaged in their learning by actively seeking and watching relevant content on YouTube. This, in turn, facilitates a better understanding of the topic. Additionally, YouTube's active engagement fosters nursing students' participation in the learning process, encouraging them to approach problems from a holistic standpoint, discuss, and analyze them, and enhance their critical thinking [7]. Moreover, YouTube plays a significant role in the field of performing arts. Utilizing YouTube as a learning tool in performing arts can adapt to current trends and encourage social collaboration within the course. This approach can ignite creativity and imagination in both students and professors, thus enhancing the overall learning experience [8]. Overall, it can be suggested that YouTube and the visual stimuli that it offers can be a powerful tool for reshaping the educational process.

### 1.1. Related Work

There are several recent studies that explore sentiment analysis methods on social media extracted datasets, e.g., [9–11]. However, most of them are not oriented in educational videos.

Recently, Anastasiou et al. in [12] applied a hybrid approach using lexicon and machine learning approaches to predict sentiment from YouTube extracted datasets. Their comparative findings demonstrated the prevalence of TextBlob over VADER to detect sentiment from YouTube comments on healthcare-related videos. Suhasini et al. [13] employed supervised learning to detect emotions in Twitter data. They compared two algorithms, namely K-nearest neighbor (KNN) and naive Bayes (NB). Their research findings indicate that naive Bayes outperformed K-nearest neighbor in this context. In another study by Jayakody et al. [14], Twitter data related to product reviews were gathered and subsequently analyzed. They utilized the support vector machine (SVM), logistic regression, and K-nearest neighbor machine learning algorithms. Additionally, they employed count vectorization and term frequency-inverse document frequency mechanisms to convert text into vectors suitable for input into the machine learning model. The highest accuracy score of 88.26% was achieved by logistic regression in combination with a count vectorizer. Furthermore, Bhagat et al. [15] adopted a hybrid approach, combining naive Bayes and K-nearest neighbor algorithms, to categorize tweets into three classes: positive, negative, and neutral sentiment. Their approach yielded superior accuracy compared to the random forest algorithm. As regards deep learning approaches, several researchers (e.g., [11,16,17]) applied BERT on text classification, achiving high perfoamnce rates of almost 92%.

Topic clustering on social media extracted datasets has been mainly performed through Latent Dirichlet Allocation (LDA) to identify topics of interest and different audiences. For instance, recently, Wahid et al. [18] applied LDA for the annotation of textual data to detect the most dominant topics on an unstructured social media dataset and then classified them through BERT embeddings. In another study, Zhang et al. [19] appled LDA to identify audience groups for books in social media.

Although numerous researchers have conducted analyses on the educational applications of social media, only a few have focused on YouTube's educational comments, with medicine being the most common domain.

Lee et al. in [20] conducted an experiment using 150 videos and gathered 29,386 comments extracted from YouTube channels to investigate whether YouTube, as a social media platform, could contribute to Self-Directed Learning (SDL). The aim of their research was to demonstrate that social technologies, such as social media platforms, could truly enhance autonomous learning by providing appropriate instructions to learners. They employed a combination of sentiment and qualitative analysis, resulting in data triangulation to emphasize the significance of the results. The sentiment analysis results indicated that out of the 150 videos, only 8 had more negative than positive comments. The main conclusions

of the study were that employing various analytical techniques to study comments/reviews on social media can lead to a better understanding of online expressions. Additionally, the study shed light on the correlation between comments and SDL traits. Furthermore, it revealed that social learning is enhanced through comments, as learners exhibited behaviors, such as sharing goals, expressing thankfulness, having fun, and displaying a more social and extroverted behavior. The practical aspects of the study were also mentioned by the researchers. Firstly, the collaboration between learners and contributors encourages SDL and participation. Secondly, it provides a powerful tool for institutions, particularly for lifelong education, by empowering the educational process.

Dubovi and Tabak in [21] also conducted research on YouTube by analyzing 1560 comments from six post-videos in the field of science. The primary goal of this study was to investigate whether social media, especially YouTube, could establish a foundation for collaborative knowledge through discourse moves. The researchers categorized their work into three segments: discourse moves, knowledge construction, and the combination of both, and they performed different types of analysis within each category. In the first category, their analysis of the comments revealed that the dominant interaction was neutral assertions or countering assertions without immediate disagreements. Subsequently, an ANOVA analysis on user tendencies showed that counter assertions and disagreements were followed by evidence. However, only a small percentage of comments supporting counter assertions or disagreements was accompanied by evidence credibility (21% of the comments). For the second category, they applied collaborative knowledge construction analysis.

Tolkach and Pratt in [22] conducted research that focused on an unexplored aspect of YouTube, namely its potential for learning in the tourism sector. The study analyzed videos from a YouTube channel called "Travel Professors", which featured short videos filmed at various locations and provided abundant research and learning material related to the tourism domain. The researchers used the YouTube analytics tool to extract data about the channel, and for the comments, they performed content analysis. The findings revealed that videos featuring less popular tourism locations garnered significant attention from viewers. Additionally, the study highlighted that students appreciated the use of videos as study and discussion material in class. Videos served as a bridge between theoretical knowledge and real-world experiences, enhancing the learning process. The true essence of combining YouTube with tourism and hospitality learning lies in approaching this domain from both theoretical and practical perspectives.

Azer et al. in [23] focused on evaluating the quality of videos on YouTube related to ileostomy and colostomy. They identified 1816 videos, of which 149 met the inclusion criteria aligned with medical instructions and guidelines from reputable medical organizations. After assessing these 149 videos, only 52 were found to have an educational purpose and presented accurate information about ileostomy and colostomy. The excluded videos lacked scientific information and often failed to adhere to proper hygiene standards. The study emphasized the importance of videos created by medical experts, organizations, institutions, or patients who had experience with these medical conditions. Such videos could significantly contribute to public education about ileostomy and colostomy.

King and McCashin (cite [24]) conducted an exploration of YouTube vlogs centered around Borderline Personality Disorder (BPD) and performed a thematic analysis of the accompanying comments. The vlogs focused on individuals living with BPD and were sourced from YouTube Ireland, utilizing the search term "Living with Borderline Personality Disorder Vlog". Through the application of rigorous inclusion and exclusion criteria, the researchers included only four vlogs for analysis. The investigation led to the identification of five key themes within the comments: "Sharing advice, support, and encouragement", "Vlogs destigmatizing, providing information, and educating", "Solidarity, relatability, and personal connection", "Intense, unstable intrapersonal and interpersonal functioning", and "Prompting disclosures about mental health struggles". Notably, the study emphasized the existence of a supportive network established by both vloggers and commenters,

particularly in response to negative emotions like depression and loneliness. The YouTube vlogs played a crucial role in promoting awareness and comprehension of BPD.

### 1.2. Research Objectives

Motivated by the above, this study aims to explore the learners' sentiment while watching educational videos created by a popular YouTube channel focused on educational content. The sentimental aspect will be assessed by analyzing large datasets of user comments. These comments will be categorized as positive, negative, or neutral, allowing for an evaluation of the videos' impact on the learning experience. Additionally, the study aims to identify thematic topics to further classify the comments into specific divisions.

The main research questions (RQ) are:

- RQ1: Which is the learners' dominant sentiment for the educational YouTube videos?
- RQ2: What are the main topics and categories of comments for the educational YouTube videos?

The findings of this study contribute to the fields of sentiment analysis, educational technology, and pedagogy by examining sentiment in the context of YouTube educational content, comparing analysis approaches, and emphasizing the positive impact of social media platforms on learning. It also provides practical recommendations for educational institutions to improve their teaching methods. The findings will also contribute to a better comprehension of the emotions evoked in people when they view educational content on YouTube. Understanding these emotions will facilitate the design of more tailored and personalized learning experiences according to learners' preferences. Additionally, the study aims to identify the elements of the educational videos that were most favored by the audience.

The study is structured as follows: Section 2 describes the methods and tools that were used to conduct this study and answer the research questions. Section 3 presents the results and Section 4 discusses the contribution and limitations of the study. Finally, Section 5 concludes with the key findings and discusses the limitations of this as well as future work that could be conducted.

## 2. Materials and Methods

### 2.1. Video Selection

Four playlists were selected from the YouTube channel 'Ted ed' to download the associated comments. The first playlist, titled 'A Bug's Life', comprises educational videos related to bugs. The second playlist is called 'Epic Engineering: How Things Get Made'. The third playlist is 'Creative Writing Workshop #CampYouTube #WithMe',

and the final one is 'Mental Health Awareness'. These playlists are required to adhere to specific criteria, as outlined in Table 1.

**Table 1.** Inclusion and exclusion criteria for the playlist selection.

| Inclusion Criteria | Exclusion Criteria |
|---|---|
| Public (the videos which are included) | Private (the videos which are included) |
| Comments and ratings enabled | Comments and ratings disabled |
| English language | No English language |
| More than 100 comments | Less than 100 comments |

The educational playlists were selected according to the above criteria, and no content/topic filter was applied. This allowed extracting comments from various audiences engaged with the content.

### 2.2. Data Extraction and Processing

The YouTube Data API and Google Sheets and Scripts were used to extract the YouTube comments. Google Apps script is a cloud-based scripting language that allows Google

Apps to improve their functionality. After retrieving the activation of YouTube Data API through Google Developers Console, a total of 167,987 comments were extracted from 59 educational YouTube videos of four lists in TedEd channel. The videos' educational topics included mental health, creative writing, building construction and insects.

After collecting the data, the next step involves cleaning it for analysis. To accomplish this, we employed a stop words function. This function enabled the algorithm to eliminate non-relevant words from the text that do not contribute any meaningful value to the sentences. Additionally, we removed characters that could introduce noise, such as "https", "@", "#", and so forth. Proceeding further, we defined the comment preproc function to handle the task of cleaning the data, specifically removing links and other characters that only consume time during analysis. Ultimately, we return the cleaned values from this function and store them in a new column to present the cleaned text.

### 2.3. Sentiment Analysis Tools

VADER (https://github.com/cjhutto/vaderSentiment, accessed on 10 April 2023) and TextBlob (https://textblob.readthedocs.io/en/dev/, accessed on 10 April 2023) lexicons were applied through python scripts to conduct the sentiment analysis task on the extracted datasets. VADER is the abbreviation of the Valence Aware Dictionary and Sentiment Resonance. VADER belongs to the lexicon-based sentiment analyses approaches that are widely spread in order to analyze comments, tweets reviews and every type of published text. A benefit that is provided from VADER is the lack of a training model, and also, it can manage difficult situations in sentences like emoticons, abbreviations, slang, etc.

The VADER analysis is concentrated around the intensity of emotions or the empathy when someone express them, which is the logic behind the VADER algorithm, resulting in higher scores from the existence of punctuation and exclamation points. The algorithm can score differently sentences that are even the same with the condition of utilizing the capital letters: for example, considering "WONDERFUL" and "wonderful", the capital word will have a higher score than the other one, even if the words are identical. The algorithm function of VADER scores each text with negative, positive, neutral and compound polarities. The values of the neutral, positive and negative are labeled between 0 and 1; however, the compound polarity is labeled between −1 and 1.

We utilized the Rake (Rapid Automatic Keyword Extraction) [25] algorithm to extract keywords and phrases from a given text. The algorithm operates in a straightforward manner, calculating the frequency of words that appear in the text. However, it excludes words with no lexical meaning, such as stop words or words containing punctuation. Another key functionality of Rake is the generation of a list containing stop words and delimiter phrases. This list is employed to segment a large text into keywords, which are then used to compute their respective frequencies within the specific text.

TextBlob is a library for data analysis, especially for sentiment processing of a text, including translation and classification. The technology used is the natural processing language (NLP). For this study, TextBlob was used to categorize the sentiment comments into positive, negative and neutral. In addition, TextBlob results will be compared with the VADER analysis results.

The algorithm has two attributes, polarity and subjectivity, which are essential for the analysis of the data. The values that polarity can take lie in the interval $[−1, 1]$, while value 0 classifies the comment as neutral. If the range of the values of polarity belongs to $(0, 1]$, then the comment is classified as positive; otherwise, it is characterized as negative. The subjectivity score has the same range as the polarity score which is $[−1, 1]$, which the objectivity is described as the value 0. TextBlob begins importing the appropriate library; then, the entire database is running assigning the polarity scores. The polarity scores need to be categorized as positive, neutral and negative. The final step is to recognize the subjectivity and save data into a CSV file.

### 2.4. Topic Modeling Using LDA

A topic modeling approach was applied on the dataset to discern overarching categories and topics related to the comments. The method employed for this analysis was Latent Dirichlet Allocation, commonly known as LDA, which falls under the category of probabilistic topic modeling. The study conducted by Abdelrazek et al. in [26] demonstrates that this process begins with creating bags of words, which are represented as vectors with a length of L. Subsequently, they propose a hypothetical statement: each document $\iota$

can generate words one at a time by randomly selecting topics from the distribution of document-specific topics, denoted as $\theta i$, and then choosing a word at random from the selected topic.

To perform the Latent Dirichlet Allocation (LDA) analysis, we applied the RapidMiner (https://rapidminer.com/, accessed on 15 May 2023) software and utilized the Parallel-TopicModel from the Mallet library as described in the work by Newman et al. [27]. The analysis employed the SparseLDA sampling scheme and corresponding data structure as proposed by Yao, Mimno, and McCallum [28]. For the sampling method, the Gibbs Sampling technique was employed.

To effectively handle the abundance of words in our dataset, specific parameters were configured. We specified our desire for the extraction of 20 distinct topics, which were each characterized by the top 10 words associated with it.

The process of implementing LDA involved the following steps:

- Data Loading: Initially, the operator selected the appropriate data source, in this case, a CSV file containing the dataset.
- Data Transformation: To facilitate the subsequent steps of the analysis, the data required transformation. To achieve this, the operator employed the "text to nominal" operator, enabling the machine to interpret and process the data accurately. This transformation also included performing necessary calculations to prepare the data.
- Attribute Selection: The operator identified and selected the relevant attributes from the dataset. In this context, the chosen attributes corresponded to the columns containing comments, as these were the segments of interest for the LDA topic modeling.
- LDA Application: With the data preprocessed and the relevant attributes selected, the LDA analysis was applied. Specifically, two LDA operators were utilized: "Extract Topics from Data (LDA)". These operators were configured to generate a total of 20 distinct topics with each topic being represented by the top 10 associated words.

### 3. Results

#### 3.1. Sentiment Analysis

According to the VADER sentiment analysis (Figure 1), the majority of comments (79.61% or 75,196 comments) have been categorized as neutral. The second most prominent emotion identified by the VADER analysis is positivity, accounting for 15.04% with 14,207 comments. Lastly, negative sentiment constitutes 5.35%, representing 5035 comments. While the dominant sentiment appears to be neutral, it is noteworthy that the number of positive sentiments surpasses the number of negative sentiments This observation suggests that the educational videos have a positive impact on the viewers. This finding aligns with the research of Lee et al. in [20], which demonstrates that a positive attitude can foster a conducive learning culture and contribute to the promotion of self-directed learning.

The TextBlob analysis (Figure 2) exhibited a similar pattern, with the majority of comments expressing a neutral sentiment, accounting for 60.22% (56,888 comments). Following that, the positive sentiment constituted 29.35% (27,722 comments), and the negative sentiment came in last with 10.43% (9851 comments) it can be inferred that despite an overall neutral sentiment, there was a noticeable presence of positive emotions in the comments, reinforcing the outcomes of the VADER analysis.

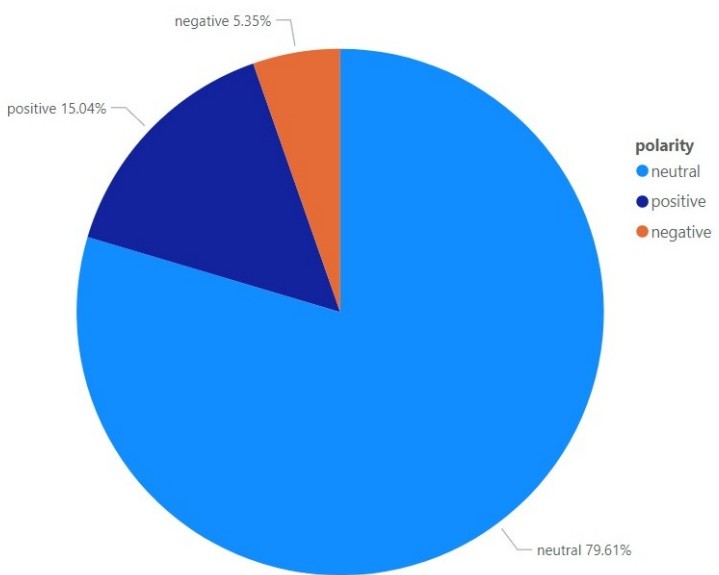

**Figure 1.** Polarity score chart pie for VADER analysis.

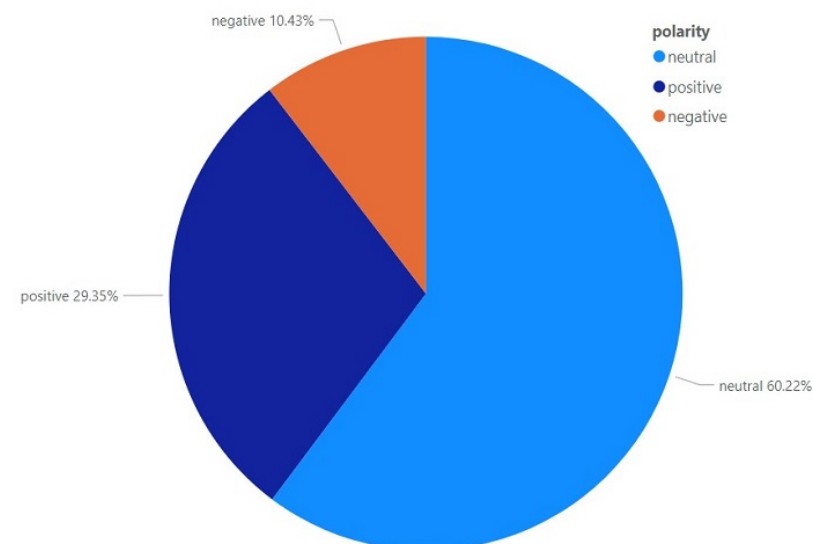

**Figure 2.** Polarity score chart pie for TextBlob analysis.

However, we can observe that the two algorithms that have been used have some differences in percentages. The VADER analysis presents high rankings in the neutral results, while TextBlob scores higher in the positive and negative sentiments. We observe that neutral segment reduced by almost 17%, while the negative and the positive are almost doubled. This phenomenon can be elucidated by the limitation of TextBlob, where numerous comments can be categorized as neutral due to its inability to recognize irony and slang words [29]. Moreover, this disparity in categorization arises because VADER demonstrates a superior capability to analyze text from social media compared to TextBlob [30].

Another intriguing aspect pertains to the evolution of emotions over time with respect to comments. Figure 3 displays three colored lines representing the sentiments: positive (Sum of pos), negative (Sum of neg), and neutral (Sum of neu). Notably, the neutral sentiment exhibits a high polarity (dark blue) throughout the graph, which is followed by the positive (orange) sentiment, and finally, the negative (azure) sentiment. An interesting

observation is that both positive and neutral sentiments follow a similar trajectory of increase and decrease.

The primary peak for all sentiments occurred in 2020, after which a decline began, spanning from 2020 to 2021, with a relatively slow decrease rate. However, post-2021, the percentage of sentiments has been on the rise. Consequently, we are left curious about what the future holds in this regard. Due to the declining trend in sentiments shown in the graph, we lack substantial evidence to determine the key factors driving these changes. Nonetheless, we assume that the onset of the declining path coincides with the period when humanity faced the COVID-19 pandemic. Further investigation is required to ascertain which elements have had such a drastic effect on TedEd videos.

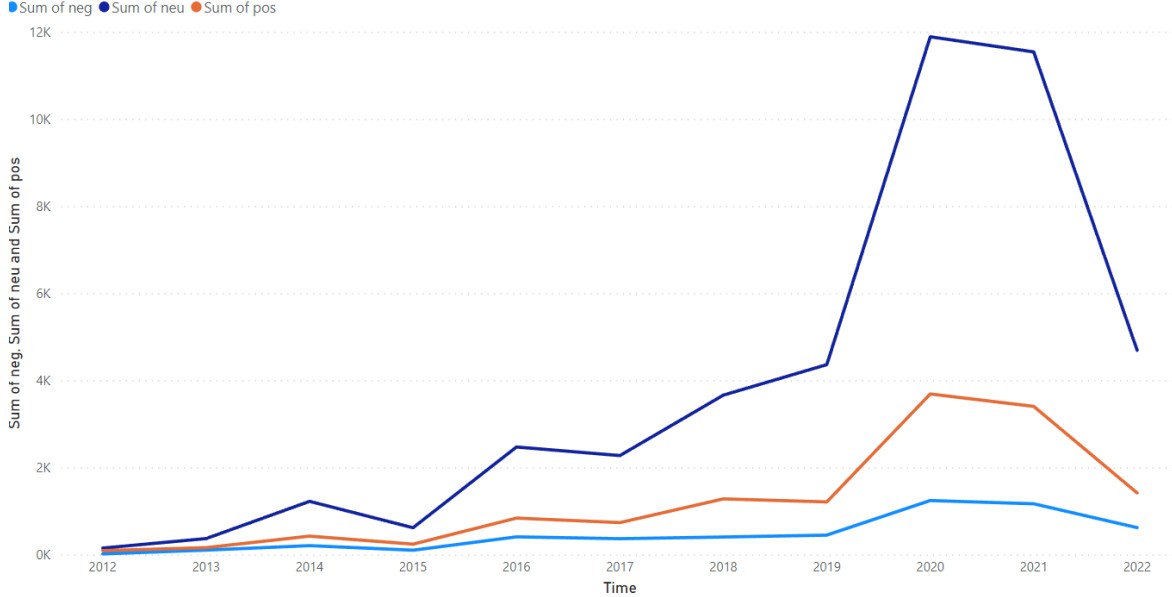

**Figure 3.** Timeline of sentiment evolution.

Topic Clustering Using LDA

Utilizing the RapidMiner processes to apply LDA on the extracted and cleaned dataset, 20 topics were successfully created. Each topic has been characterized by the words contained in it. For example, topic 0 refers to "The Lord of the Rings", which belongs to creative writing, while topic 1 is about how much the viewers liked the videos of TedEd. Then, these 20 topics were categorized into five categories based on the words they include. The five categories are as follows:

- Bugs;
- Construction;
- Creative writing;
- Mental health;
- Video appraisal.

One of the most interesting topics focuses on words with similar meanings that express fondness for the TedEd videos. During this procedure, we faced some issues because the language used was informal, and some words were in other languages. Nevertheless, the analysis attempted to categorize even those words. Figures 4 and 5 depict the weighted words and the words assigned to each topic. The LDA analysis concluded that the words "video" and "quots" have an influence on the topics. These words are located at the center with the biggest bubbles, indicating a higher frequency of appearance. Additionally, positive words like "like" and "love" hold significant importance. Moreover, there are positive or neutral words suggesting that the educational videos were very interesting and created a positive attitude among learners. Another noteworthy point is that certain words appeared more times than others, but this could be explained by different topics including

the same words. Furthermore, from a human perspective, an effort has been made to assign names to each topic and categorize them into themes. Figure 5 presents the named topics and their corresponding themes.

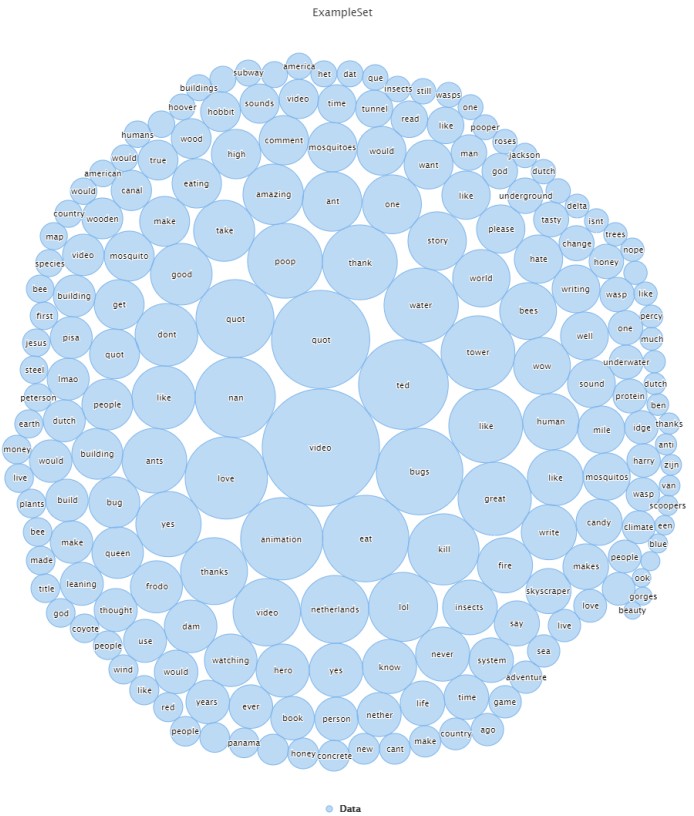

**Figure 4.** Bubble chart with the weighted words.

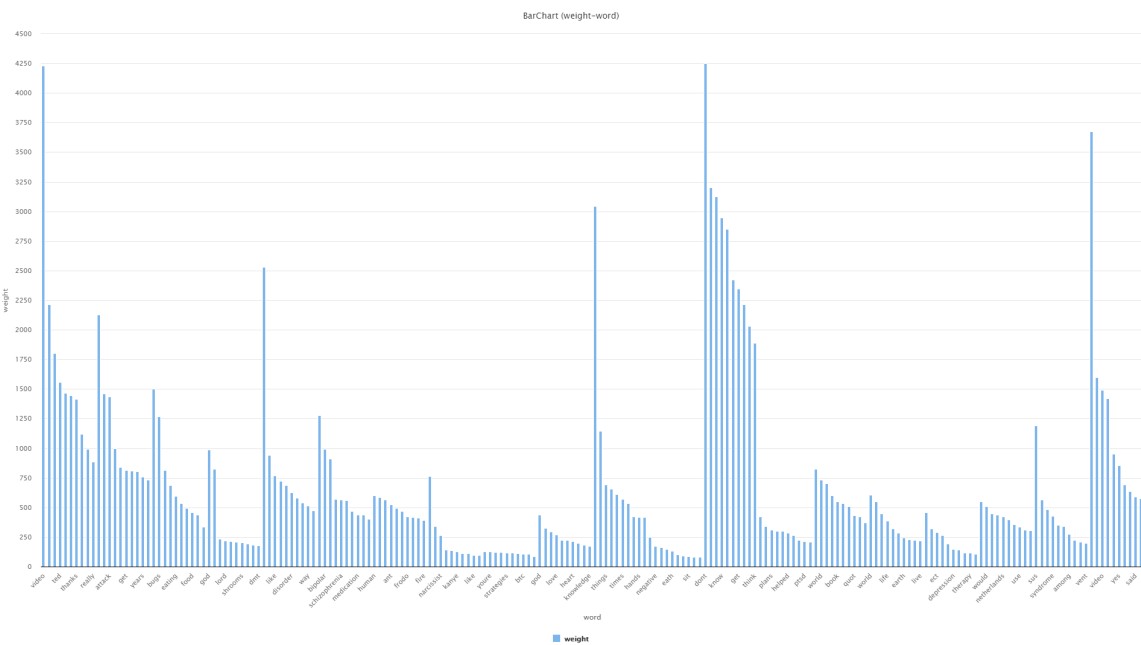

**Figure 5.** Bar plot of the weighted word for each topic.

There is a big concentration of words for the topics that refer to the video's appraisal main theme and the subtopics. Regarding the results of the sentiment analysis, individuals

found the educational videos with the specific traits very interesting. In addition, the topics of this category suggest that the video animation combined with the sound it creates an atmosphere in the mind of the learner, which reflects that time passes smoothly watching the video.

## 4. Discussion

### 4.1. Principal Findings

The sentiment analysis results indicated that the majority of the comments collected from YouTube were categorized as neutral. TextBlob and VADER generated similar results, revealing though some differences regarding the percentage of neutral versus positive and negative outcomes. This outcome aligns with previous findings that compared TextBlob and VADER lexicons to identify sentiment from YouTube comments [12]. The differences between the two lexicons can be explained by their differences in recognizing some types of words expressed in social media [29,30]. However, in the current study, there is a noticeable inclination for viewers to express positive comments rather than negative ones. Since the percentage of negative comments was quite low, it is evident that the neutral and positive sentiments follow a similar pattern, which refers to the way experiences increase and decrease. This suggests a potential hidden relationship between neutral and positive comments that warrants further investigation.

The LDA analysis aimed at exploring the presence of topics and themes, and the results revealed the existence of 5 themes among 20 topics. It should be noted that the selection of these topics was made randomly. Additionally, the LDA analysis highlighted certain elements that significantly influence learners' perception of the videos, such as animation, music, and the messages conveyed. These elements play a crucial role in fostering a positive attitude among learners and may even influence their behavior toward becoming more outgoing [20]. Furthermore, in alignment with previous studies (e.g., [22,23]), the topic clustering results emphasized the utility of educational videos on learners' interest and engagement.

### 4.2. Contribution and Practical Implications

The research study provided has several notable research contributions:

Sentiment Analysis of YouTube Comments: The study contributes to the field of sentiment analysis by analyzing comments made by YouTube learners on educational content. This is significant because sentiment analysis is typically applied to a wide range of texts, and this study specifically focuses on educational content, shedding light on the sentiment of learners in this context.

Comparison of Sentiment Analysis Approaches: The study compares two popular lexicon-based approaches, TextBlob and VADER, to analyze sentiment. This comparison provides insights into the effectiveness and differences between these methods when applied to YouTube comments, adding to the methodology of sentiment analysis.

Positive Attitude toward Educational Videos: The study's findings reveal that learners exhibited a positive attitude toward the educational videos on YouTube. This insight is valuable for educators and content creators, as it highlights the effectiveness of using platforms like YouTube to create a positive learning environment.

Topic Identification Using LDA: The study goes beyond sentiment analysis by employing Latent Dirichlet Allocation (LDA) to identify the main topics of interest to viewers. This adds depth to the analysis by understanding what specific elements or information in the videos attract learners' attention. This can inform content creators about the preferences of their audience.

Educational Contribution of Social Media: The research emphasizes the significant contribution of social media platforms, particularly YouTube, to the educational process. This is a noteworthy finding as it highlights the role of these platforms in facilitating learning and suggests that educational institutions can leverage this to enhance the learning experience.

Recommendations for Educational Institutions: The study concludes by suggesting that educational institutions can benefit from the insights gained in the research. By creating subject-specific videos as supplementary material and adapting to students' preferences, institutions can enhance the overall learning experience. This recommendation has practical implications for educators and institutions seeking to improve their teaching methods.

### 4.3. Limitations and Future Research

The limitations that emerged during the research could potentially impact the research outcomes. These limitations are evident from the initial step of data retrieval. Specifically, when choosing the YouTube platform as the primary source for the research, we disregarded other platforms, such as Instagram, which also provide educational content through short-duration videos aimed at offering valuable information to individuals. Consequently, the sentiment analysis of YouTube videos may not accurately represent the sentiments of the entire online community regarding educational videos. Another challenge we encountered was determining the appropriate number of videos to analyze. We had to consider whether to use various videos from different channels or stick to a single channel.

Additionally, we questioned whether the selected videos should share similar characteristics or not. The decisions made regarding these aspects, as well as the chosen research path, could potentially yield different results and findings. Similarly, the video list was not based on similar audience characteristics and interest topics; hence, future research could potentially examine sentiment among viewers of similar characteristics for a selected list of educational YouTube videos. To proceed with this study, we had to focus on YouTube comments written in English, which posed a significant limitation since it led to the exclusion of certain comments. Despite applying proper data-cleaning techniques, some non-English words persisted, particularly in topic 17. These words, even after cleaning, were somehow categorized by RapidMiner, implying that they still held meaning within the topic. Moreover, the algorithms utilized for sentiment analysis, namely TextBlob and VADER, did not offer a definitive depiction of the overall sentiment, even though the topic clustering suggested a positive tone in the videos. Consequently, we resorted to using both algorithms as a means of confirming the results. By doing so, we could verify the outcomes even if there were slight deviations between the methods. To ensure the validity of the results, we employed three different tools, considering the limited existing research in this particular field, and we aimed to utilize all available resources to guarantee the validity of our findings.

A potential avenue for future research could involve conducting a comparative analysis between videos on YouTube and Instagram. Such an analysis would aim to highlight the disparities between these two social media platforms and explore how learners are impacted by their differences. Additionally, the analysis reveals an intriguing observation regarding the undefined relationship that emerges between neutral and positive sentiments. Further investigation is required to determine whether this occurrence is merely a chance happening within the provided data or if there exists a meaningful connection between these two variables. To address this, a larger sample of videos sharing similar characteristics should be examined to ascertain the validity of this relationship and whether it indeed holds true. It would be also essential to examine the relationship between the detected positive and neutral comments as well as to examine the performance of LDA on different social media platforms. Finally, additional analyses on different and larger datasets would reinforce the findings and provide deeper insights.

## 5. Conclusions

This study aimed to investigate the sentimental polarity of comments made by YouTube learners on channels with educational content. The data were collected from a popular channel called TedEd, which offers a wide range of educational videos presented by experts in various fields. We employed two widely used lexicon-based sentiment analysis methods, TextBlob and VADER, for comparative analysis. While these methods

exhibited variances in their categorization of words as neutral, positive, or negative, they consistently revealed a prevailing positive sentiment, albeit with neutrality being the most frequent emotional state. In summary, our sentiment analysis affirms that learners possess a predominantly positive attitude toward these educational videos, thereby cultivating a constructive learning atmosphere.

Furthermore, we harnessed Latent Dirichlet Allocation (LDA) to unearth the core topics of interest for viewers, illuminating the aspects and information within the videos that captivated their attention and preferences.

Our primary discovery underscores the profound educational potential harbored within social media platforms, notably YouTube. In essence, our findings highlight the capacity of social media platforms, such as YouTube, and educational channels to nurture a favorable learning milieu through their abundant and varied content. Educational institutions can harness this insight by integrating the distinctive attributes of these videos into their pedagogical approaches. By creating subject-specific videos as supplementary resources, educational institutions can enrich the learning journey and align with the inclinations of their student body. Limitations of the study regard the characteristics of the selected videos and audience characteristics, while future work can be conducted on different datasets to reinforce the results and provide deeper insights.

**Author Contributions:** Conceptualization, I.C. and K.T.; Methodology, I.C., K.T. and D.K.; Software, D.K.; Validation, D.K. and C.T.; Resources, D.K.; Data curation, I.C.; Writing—original draft, I.C.; Writing—review & editing, K.T., D.K. and C.T.; Visualization, I.C.; Supervision, K.T.; Project administration, K.T. and C.T.; Funding acquisition, C.T. All authors have read and agreed to the published version of the manuscript.

**Funding:** This research is co-financed by Greece and the European Union (European Social Fund-SF) through the Operational Programme "Human Resources Development, Education and Lifelong Learning 2014-2020" in the context of the project "Support for International Actions of the International Hellenic University", (MIS 5154651).

**Data Availability Statement:** The data presented in this study are available on request from the corresponding author. The data are not publicly available due to working on more research analyses.

**Conflicts of Interest:** The authors declare no conflict of interest.

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
