# Peer review of "Learning Analytics on YouTube Educational Videos: Exploring Sentiment Analysis Methods and Topic Clustering"

_electronics, doi:10.3390/electronics12183949_

Round 1
Reviewer 1 Report
1- After reading the abstract, the condensed and clear research findings were missing, thus, this is a less significant contribution to the body of literature.
2-Authors selected four video contents but each of them have different focus and content, it is not clear to me how the author guarantees that the audience has the same interest because different audiences have different perspectives and preferences.
3-a total of 167.987 comments were received, I was confused "is that 167,987 reviews or one hundred and sixty seven"
4-the data cleaning process is not detailed. How the authors do the cleaning with comments that include symbols, not completed words, words with lowercase or capital and the stem word ... this makes your data not reliable...
5-LDA topic is not clearly explained; for example, what technique of Parallel Topic applied? LDA implementation was missing, again go to LDA sampling approach also not mentioned.
Author Response
We would like to thank the reviewer for their valuable comments, which have significantly contributed to the improvement of the quality of our paper.
- In response to the reviewer's feedback, we have made revisions to the Abstract. We have customized and extended the text to provide a clearer demonstration of our research findings and the contribution of our paper.
- Building upon the reviewer's insightful comments, we have incorporated additional text in section 2.1 (Video Selection) to provide a more explicit explanation of our video selection criteria. This clarifies that our selection was based on generic criteria and did not necessitate a similar target audience, as different topics of interest were considered. Furthermore, we have included supplementary content in section 4.1 (Limitations and Future Research) to suggest potential directions for future research.
- We appreciate the reviewer's keen eye for detail in pointing out a typographical error. We have rectified this issue by correcting the symbol to accurately represent thousands rather than hundreds.
- As suggested, we have expanded the discussion of data cleaning and processing in section 2.2. Additionally, we have renamed the section to "Data Extraction and Processing" for improved clarity.
- In accordance with the reviewer's advice, we have provided more comprehensive explanations in section 2.1 to elucidate the LDA (Latent Dirichlet Allocation) analysis. This added text offers a more concrete understanding of our approach.
Reviewer 2 Report
This study examines YouTube's rising popularity as a communication bridge, focusing on its educational content. Analyzing 167.987 comments from educational channels, VADER and TextBlob sentiment analysis suggests predominant neutral and positive sentiments, with Latent Dirichlet Allocation revealing diverse thematic clusters. Followings are my concerns:
1. Abstract requires enhancement to effectively showcase the accomplished research work.
2. Recent literature can be examined to underscore the significance of the research topic through a comprehensive review of current works. For example, https://doi.org/10.3390/s23010506.
3. The clarity of illustrating the contributions made by this work could be improved for better understanding and recognition.
4. Enhancement is needed for Figures 3 and 5 to be more effective and informative.
5. Anticipated are additional experimental findings along with thorough analyses to provide greater depth and insight.
Minor editing of English language required
Author Response
We would like to thank the reviewer for their valuable comments.
- As suggested, we have optimized the text in Abstract to effectively showcase the accomplished research work.
- We enriched the literature with several recent works including the one suggested by the reviewer.
- Based on the reviewer’s useful feedback, the contribution of our paper has been more clearly demonstrated as we enriched the Discussion section by adding one new subsection ‘Practical Implications, and we provided some more evidence in Introduction (1.2) and Abstract.
- We added some more text in section 4.1 to suggest future work for additional experimental analysis.
Finally, we optimized the English language in the text.
Reviewer 3 Report
Thank you for the opportunity of reviewing very interesting article Learning Analytics on YouTube Educational Videos: Exploring Sentiment Analysis Methods and Topic Clustering. In this study autors explore the sentiments of YouTube learners by analyzing their comments on educational YouTube videos. The empirical basis of the study was 167.987 comments were extracted and processed from educational YouTube channels through the YouTube Data API and Google Sheets. Lexicon-based sentiment analysis was conducted using two different methods: VADER and TextBlob.This approach seems to be extremely relevant and promising.
Comments and Suggestions for Authors:
• Sentiment analysis is one of the most developed areas in the field of natural language processing. Authors should be encouraged to review the mainstream work in this area in more detail.
• The empirical basis of the study was 167,987 comments extracted and processed from educational YouTube channels using the YouTube data API and Google Sheets, which seems to be an insufficient number. Expanding the database will give greater credibility to the conclusions drawn by the authors. Maybe. this will be done in further research.
• The research topic seems to be very relevant, since the distribution of educational content on YouTube has fundamentally changed the educational environment. Comparison with other digital playgrounds would be very useful from a methodological point of view.
• Please describe in detail how your study fits for aims and scope of Special Issue Big Data and Large-Scale Data Processing Applications (Electronics).
• For theoretical framework and bibliography additional current references should be included to new research 2022-2023.
• Please adjust the structure of the article according to the requirements.
• The process of discussing the results can be extended by applying the results and extrapolating them to other similar studies.
Author Response
First, we would like to thank the reviewer for their useful feedback.
We extended the literature review by adding several recent advances in the field of SA on social media datasets, and provided more details as suggested by the reviewer. Also, we added the suggestion of a larger dataset as a future task. Overall, we included more recent references in the bibliography, and we extended the discussion of the paper to include implications and comparison with relevant studies.
Reviewer 4 Report
The purpose of this manuscript is to explore the sentiments of YouTube learners by analyzing their comments on educational YouTube videos. The data was collected from TedEd, a channel which offers educational videos presented by experts in various fields. TextBlob and VADER were applied to compare the results. The analysis revealed that learners exhibited a positive attitude towards educational videos, „creating a positive learning environment”.
In addition, the authors used LDA to identify the main topics of interest and understand the elements in the videos that attracted the preference of the viewers. They concluded that YouTube can contribute to the educational process.
In my opinion, the paper addresses a very interesting and topical issue, with a very interesting analysis.
However, from my point of view, the authors must develop the conclusion section. In addition, they should identify other current bibliographic sources and report their results to these sources.
Overall, I evaluate the study very positively and I recommend its publication after minor revisions.
Author Response
We would like to thank the reviewer for appreciating our work.
According to his/her feedback we have optimized the Conclusion section and we added some more sources to report our results.
Reviewer 5 Report
This interesting study aimed to present using sentiment analysis methods and topic clustering on the YouTube educational videos by learning analytics. The authors in this study focused to explore the sentiments of YouTube learners by analysing their comments on educational YouTube videos. The authors of this work aimed to explore, based on polarity of the dominant sentiment, the efficiency of educational YouTube videos.
Experiment results show that the dominant sentiment expressed in the comments was neutral, followed by positive sentiment, while negative sentiment was the least common. Regarding and the results of the sentiment analysis, individual found the educational videos with the specific traits very interesting. The topics of this category suggest that the video animation combining with the sound, it creates an atmosphere on the mind of the learner, that reflects that time passes smoothly watching the video.
The authors evaluated videos which were selected from the YouTube channel ’Ted ed’. The sample consist of four playlists and their download, the associated comments. A total of 167,987 comments were extracted and processed from educational YouTube channels through the YouTube Data API and Google Sheets. I consider the sample size like sufficient.
The measurements and instruments used by the authors seem to be valid. The experimental results are processed in detail, with many calculations and table confirmation of results.
The discussion reveals that the sentiment analysis results indicated that most of the comments collected from YouTube were categorized as neutral. In the current study, there is a noticeable inclination for viewers to express positive comments rather than negative ones. The authors suggest that exist a potential hidden relationship between neutral and positive comments. The discussion summarizes the essential findings of the research. The measurements and instruments used by the authors seem to be valid. The results are processed in detail and graphically. More literature can be added to the discussion, enriching the authors' arguments.
Do the authors plan another relationship between neutral and positive comments in videos in the future? What will be the next direction of research?
Limitations of the research are part of the article. I agree with the limitations of the research.
Are there restrictions when it will not be possible to use the presented sentiment analysis on the different platform like YouTube?
The paper I evaluate positively because the presented study identifies on the base of the LDA analysis highlighted certain elements that significantly influence learners’ perception of the videos, such as animation, music, and the messages conveyed. These elements play a crucial role in fostering a positive attitude among learners and may even influence their behaviour.
Author Response
We would like to thank the reviewer for appreciating our work.
According to his/her feedback we have enriched the discussion including relevant references.
We also, added some text in Limitations and future research to suggest future analysis on the comments’ polarity, as well as on the application of different platforms.
Round 2
Reviewer 1 Report
Data extraction and processing has been improved a lot.
Methodology and its science processing are acceptable.
Finally, conclusion and contribution are also improved and make sense.